# Endocardial Fibroelastosis as an Independent Predictor of Atrioventricular Valve Rupture in Maternal Autoimmune Antibody Exposed Fetus: A Systematic Review with Clinicopathologic Analysis

**DOI:** 10.3390/diagnostics13081481

**Published:** 2023-04-20

**Authors:** Monika Kantilal Kotecha, Khurshid Merchant, Charmaine Jiahui Chan, Jonathan Tze Liang Choo, Krishna Revanna Gopagondanahalli, Dyan Zhewei Zhang, Teng Hong Tan, Sreekanthan Sundararaghavan

**Affiliations:** 1Department of Paediatric Subspecialties, Cardiology Service, KK Women’s and Children’s Hospital, Singapore 229899, Singapore; 2Department of Pathology and Laboratory Medicine, KK Women’s and Children’s Hospital, Singapore 229899, Singapore; 3Department of Neonatology, KK Women’s and Children’s Hospital, Singapore 229899, Singapore

**Keywords:** neonatal lupus, maternal autoantibodies, endocardial fibroelastosis, complete heart block, anti-Ro/SSA antibodies, anti La/SSB antibodies, mitral valve rupture, tricuspid valve rupture

## Abstract

Background: Neonatal lupus (NL) is a clinical syndrome that develops in the fetus as a result of maternal autoimmune antibodies. Congenital complete heart block (CHB) is the most common manifestation, while extranodal cardiac manifestations of NL, such as endocardial fibroelastosis (EFE) and myocarditis, are rare but more serious. Less is known about this atrioventricular valve rupture due to valvulitis as a consequence of maternal autoantibodies. We have described a case of cardiac neonatal lupus with an antenatally detected CHB patient who developed mitral and tricuspid valve chordal rupture at 45 days of age. We compared the cardiac histopathology and the fetal cardiac echocardiographic findings of this case with another fetus that was aborted after being antenatally diagnosed with CHB but without valvar rupture. A narrative analysis after a systematic review of the literature regarding atrioventricular valve apparatus rupture due to autoimmune etiology along with maternal characteristics, presentation, treatment, and outcome have been discussed in this article. Objectives: To describe published data on atrioventricular valve rupture in neonatal lupus, including clinical presentation, diagnostic evaluation, management, and outcomes. Methods: We conducted a PRISMA-compliant descriptive systematic examination of case reports that included accounts of lupus during pregnancy or in the newborn period that resulted in an atrioventricular valve rupture. We gathered information on the patient’s demographics, the details of the valve rupture and other comorbidities, the maternal therapy, the clinical course, and the results. We also used a standardized method to evaluate the cases’ quality. A total of 12 cases were investigated, with 11 cases drawn from 10 case reports or case series and 1 from our own experience. Results: Tricuspid valve rupture (50%) is more common than mitral valve rupture (17%). Unlike mitral valve rupture, which occurs postnatally, the timing of tricuspid valve rupture is perinatal. A total of 33% of the patients had concomitant complete heart block, while 75% of the patients had endocardial fibroelastosis on an antenatal ultrasound. Antenatal changes pertaining to endocardial fibroelastosis can be seen as early as 19 weeks of gestation. Patients with both valve ruptures generally have a poor prognosis, especially if they occur at close intervals. Conclusion: Atrioventricular valve rupture in neonatal lupus is rare. A majority of patients with valve rupture had antenatally detected endocardial fibroelastosis in the valvar apparatus. Appropriate and expedited surgical repair of ruptured atrioventricular valves is feasible and has a low mortality risk. Rupture of both atrioventricular valves occurring at close intervals carries a high mortality risk.

## 1. Introduction

Neonatal lupus (NL) affects 2% of the offspring exposed to maternal autoantibodies [1]. Complete heart block (CHB) is the most common and permanent consequence of maternal autoantibodies in the fetus. However, extranodal neonatal cardiac lupus, such as myocarditis with endocardial fibroelastosis (EFE), results in neonatal or late-onset dilated cardiomyopathy and subsequent valvar regurgitation. EFE is a non-specific, protracted response to myocardial wall stress that often worsens over time and leads to heart failure [2]. The effect of maternal autoantibodies on the endocardium, causing chordal rupture and resulting in severe regurgitation, is rarely studied, with only a few case reports or series. CHB in the context of NL is extensively studied; however, the medical literature rarely addresses the effect of myocardial involvement as well as valve pathologies in neonatal lupus. Since serious valve rupture in the neonatal period is poorly tolerated and results in hemodynamic compromise, timely repair of these valves is generally not considered. Hence, our objective was to review the medical literature and provide further insights into this less known complication of neonatal lupus. We discuss a case of an antenatally detected CHB patient with normal cardiac function who developed severe mitral and tricuspid valvar regurgitation due to chordal rupture at 3 months of age. We compare the histological findings of this case with another case who had antenatally detected CHB without valvular rupture due to maternal autoimmune disease and underwent termination of pregnancy at 22 weeks of gestation. We describe the histopathological findings of the cardiac structure in both cases. We performed a systematic literature review on tricuspid and mitral valve chordal rupture in neonatal lupus. The objective of this review is to describe the clinical, echocardiographic, and histopathological findings and outcome in patients with valvar rupture attributable to maternal autoantibodies, as well as explore any association between valvar rupture and the presence of EFE.

## 2. Case Studies

### 2.1. Case Description

#### 2.1.1. Case A

A 35-year-old gravida 2, para 1 mother was referred to our fetal cardiology center at 21 weeks of gestation for evaluation of fetal hydrops and bradycardia. An echocardiogram showed a ventricular rate of 74 bpm and an atrial rate of 144 bpm (Figure 1a). There was patchy hyperechogenicity of the atrial and ventricular walls, including the septum and chordae (Figure 1b/Appendix A). The fetus had moderate tricuspid regurgitation (TR) and mild mitral regurgitation (MR), along with pleural and pericardial effusion, and a cardiovascular profile score of 8 (Figure 1c/Appendix A).

The mother was strongly positive for anti-Ro and anti-La antibodies. Karyotype, infective screening markers, and the Kleiner test were negative. She was started on oral dexamethasone (8 mg) once daily and hydroxychloroquine (200 mg) once daily at 24 + 5 weeks of gestation. Heartrate was monitored for several weeks at the fetal cardiology center. She was started on oral terbutaline at 31 + 6 weeks’ gestation as the fetal heartrate dropped to less than 60 bpm. Her cardiovascular profile score remained between 6 and 8 throughout the pregnancy. The pleural effusions resolved by 29 + 4 weeks of gestation, and the pericardial effusion remained mild. The hyperechogenicity and valvar regurgitation remained stable throughout the pregnancy. An elective lower segment cesarean section was performed in view of increasing cardiomegaly and static growth associated with rising middle cerebral artery peak systolic velocity.

A female baby was delivered at 32 + 4 weeks of gestation with a birth weight of 1390 g and an APGAR score of 5, 9. The neonate was intubated and started on an isoprenaline infusion. CHB was confirmed by postnatal electrocardiogram (Figure 2), and echocardiography showed mild to moderate TR with no evidence of MR, and the atrioventricular valves appeared to be structurally normal (Figure 3a,b/Appendix A). There was hyperechogenicity on both ventricular myocardial walls, papillary muscles, and chordae. Cardiac function was normal (Figure 3c/Appendix A). There was minimal pericardial effusion and no pleural effusion. After stabilization of the neonate, temporary epicardial pacing wires were inserted with VVI pacing mode on day two of life due to bradycardia. The neonate was extubated to room air on day 21 of life, and weekly follow-up echocardiograms showed stable cardiac function.

However, on day 45 of life, the baby developed acute respiratory distress and lactic acidosis, requiring inotropic infusion. The septic workup was negative; however, anti-Ro titers were positive. An echocardiographic evaluation showed moderate TR and severe MR. Flail mitral and tricuspid valve leaflets were noted with ruptured chordae resulting in valve prolapse and severe regurgitation (Figure 4/Appendix A). Pleural and pericardial effusions were present with hepatic dysfunction and seizures while on steroids and inotropic supports. The neonate expired at 55 days of age.

#### 2.1.2. Case B

A 35-year-old G2P1 with no prior history of autoimmune disease was referred to our center at 20 + 4 weeks of gestation for fetal hydrops and bradycardia. CHB was detected in fetal echocardiography with an atrial rate of 112 bpm and a ventricular rate of 62 bpm (Figure 5a). Mild to moderate TR was present without MR. Biventricular hyperechogenicity of the papillary muscles and endocardium was noted with good coaptation of the valve leaflets (Figure 5b/Appendix A). Small pericardial and pleural effusions were also noted. She was extensively investigated for hydrops, and her anti-Ro antibody titers were strongly positive with negative anti-La titers. She declined medical therapy and had her pregnancy terminated. 

### 2.2. Postmortem Examination

#### 2.2.1. Gross Morphology

Case A: A cardiac postmortem showed the tricuspid valve (TV) had an abnormally small shrunken anterior leaflet with ruptured papillary muscles. The mitral valve (MV) is abnormal with dysplasia of the central portion of the anterior leaflet without the chordal attachments (Figure 6a,b). The surfaces of the myocardium and the pericardium were normal.

Case B: Pale appearance of the tips of papillary muscles over both the MV and TV without chordal rupture (Figure 6c).

#### 2.2.2. Histopathology

Case A: The histopathological examination of the cardiac tissue in Case A demonstrated calcification of the tip of the papillary muscle extending into the chordae tendineae (Figure 7a). There was rupture of the chordae of the posterior papilla with the myxoid change in MV. The atrioventricular node showed extensive calcification and fibrosis (Figure 7b). There was diffuse subepicardial calcification along with the right ventricular endocardium without fibrosis and subpericardial left ventricular calcification (Figure 7c). Normal coronary arteries were present without any signs of vasculitis (Figure 7d). Postmortem findings did not show active inflammatory cells or signs of prematurity in the lungs, liver, or brain.

Case B: The histopathological examination of the cardiac tissue showed extensive calcification and fibrosis with hypercellularity of the atrioventricular node and papillary muscles (Figure 8). Similar changes were also noted in the subendocardial portions of the left ventricle (LV) and the epicardial regions of the right ventricle (RV). Granulation tissue with multinuclear giant cells with calcification and fibrosis in the atrioventricular node and papillary muscle extending to the myocardium were observed (Figure 8).

## 3. Review of the Literature

Congenital heart block is the most common manifestation of the autoimmune effects of maternal antibodies on the fetus [1]. EFE and cardiomyopathy due to autoantibody-mediated myocarditis are discussed less frequently. Valvar regurgitation due to heart block or reduced ventricular contractility associated with EFE changes has been reported [1,3,4]. However, acute valve rupture resulting in severe regurgitation of the atrioventricular valves, which requires emergency management, is rarely discussed. The purpose of this systematic review is to summarize the current literature on atrioventricular valve rupture due to NL, including clinical presentation, diagnostic evaluation, management, and outcomes.

### 3.1. Method 

We registered this study as a systematic review protocol on Research Registry under the ID number: reviewregistry1572, with the following outlined methods.

### 3.2. Search Strategy 

We conducted a systematic search of the PubMed, Embase, Scopus, and Web of Science databases from inception to March 2023. After preliminary database searches (with databases specified), no existing or continuing mixed-method or individual systematic reviews on the subject have been found. The following search terms, which include controlled vocabulary and associated keywords, were utilized both individually and collectively: “neonatal lupus”, “atrioventricular valve”, “rupture”, “endocardial fibroelastosis”, “autopsy”, and “immunohistopathology” (refer to Appendix A). The search terms, generated by author MKK, were peer-reviewed by author SS. The reference lists of the retrieved articles were manually screened for additional relevant studies. In addition, we performed a citation search for the relevant articles.

### 3.3. Inclusion and Exclusion Criteria

We included studies that fulfilled the following criteria: (1) patients with a history of documented fetal or neonatal lupus; (2) patients with atrioventricular valve rupture; (3) articles with full text describing patient presentations; (4) human subject studies; (5) English-language or other articles with an English abstract that are being evaluated for inclusion. We excluded cases of (1) adult patients or pediatric onset of autoimmune disease, (2) reports where there was no history of atrioventricular valve rupture, (3) history of valve regurgitation attributable to myocarditis, endocardial fibroelastosis, or dilated cardiomyopathy, but where there was no mention of “rupture” in the article, (4) articles where authors determined valve rupture occurred idiopathically or by another etiology entirely, (5) studies with no case description or conference abstracts, and (6) animal or in vitro studies.

### 3.4. Study Selection

After eliminating duplicates, two reviewers (MKK and SS) independently reviewed references in two steps using an online systematic review tool (Covidence systematic review software, Veritas Health Innovation, Melbourne, Australia). Titles and abstracts were initially examined by reviewers, who eliminated those that did not satisfy the criteria for inclusion or were not available, but retained any that did not contain enough information for a full-text evaluation. Reviewers then separately assessed full-text papers, excluding any that did not fulfill the criteria for inclusion or that were unavailable.

### 3.5. Data Extraction

For each included article, MKK extracted all data that was relevant to the study. The author extracted (1) maternal characteristics (i.e., age of mother, antibody status, prior affected pregnancy) and study characteristics (i.e., author and year of publication), (2) gestational age for initial presentation, valve involvement, maternal therapy and its outcome, details of comorbidities (complete heart block, endocardial fibroelastosis, and effusion), (3) course of illness (i.e., progression of comorbidities and duration), (4) clinical interventions (i.e., antenatal, and postnatal medical and surgical therapy and its outcome), and (5) autopsy or histopathological findings.

### 3.6. Quality Assessment

Joanna Briggs Institute (JBI) critical appraisal checklists for case reports were used by one reviewer (MKK) to evaluate the quality of the case reports and case series that were included [5]. An additional reviewer (SS) double-checked this. There are eight case report questions in the tool. We classified articles as high quality if they contained reported items that exceeded two-thirds of the JBI checklist. Less than one-third were classified as low quality and excluded, and more than one-third were classified as being of moderate quality (see Appendix A).

### 3.7. Data Analysis

Due to the narrative form of this study, we used descriptive statistics to report demographic and clinical factors.

## 4. Results

### 4.1. Study and Patient Demographics

The preferred reporting items for systematic reviews (PRISMA) flow diagram [6] is shown in Figure 9. A total of 2 conference abstracts were disqualified because the full texts were not available (included in the 744 studies excluded in the PRISMA diagram). Ten of the eleven articles that were found during the original search were used in the analysis. One research article with tricuspid and mitral valve rupture in two cases was disqualified after data extraction, as the JBI checklist showed high bias due to the lack of complete data on the patient history and therapy [7] (Appendix A). The studies were mostly retrospective case series or case reports, with a total of 11 cases of atrioventricular valve rupture due to NL.

A total of 12 cases of acute valvar rupture attributable to antenatal exposure to maternal autoantibodies were analyzed (Table 1). This includes Case A from our experience (numbered as Case 1) and 11 patients extracted from 10 reports (numbered Cases 2–10) [8,9,10,11,12,13,14,15,16,17]. Shiraishi et al. reported cases of acute MR in infants, of which two cases were attributable to maternal autoimmunity [15]. The same author also reported a case of acute MR in a previous report with similar characteristics, and, hence, we have avoided duplication [13]. Another interesting report described by Brooks et al. (Case 10) observed that there were antibodies during the pregnancy with fetal heart block, while the fetus from a previous pregnancy was noted to have changes of EFE with valve rupture but without associated heart block. Hence, the authors have attributed the valve rupture to be related to undiagnosed maternal antibodies [14]. The age of the mothers ranged from 25 to 31 years. Furthermore, 42% (5/12) of mothers (Cases 2–4, 8, and 12) had known autoimmune disease [9,10,16,17], specifically Sjogren’s syndrome, affecting two mothers (Cases 3 and 4) [10,17]. Two mothers had previously affected children (Cases 3 and 7) [10,16,17] and one had a sibling that was affected subsequently (Case 10) [14]. Three mothers had normal pregnancies prior to the affected fetus (Cases 1,2, and 12) [9]. While all mothers were associated with positive Ro titers, only five (42%) had additional positive La antibodies (Cases 6 and 8–10) [12,13,14,16].

### 4.2. Comorbidities and Course

Of the 12 patients with valve rupture, 4 patients (Cases 1–4) had both MV and TV rupture (33%) [9,10,17]. Although six (50%) patients had isolated TV rupture (Cases 7–10) [8,9,11,12,14,16], only two (17%) (Cases 5 and 6) had an isolated MV rupture [13,15] (Figure 10a). Table 2 describes the antenatal echocardiographic features of these cases as reported. The changes in EFE were noted in nine (75%) patients (Cases 2–4, 7–9, and 11) antenatally [8,9,10,11,12,16,17]. These changes were noted in patients as early as 19 weeks of gestational age (GA) in three cases (Cases 2, 3, and 8) [9,10,16] and as late as 24 weeks (Case 4) [17]. All of these patients had echogenicity on the TV/MV chordae and in the papillary muscle. Two patients (Cases 1 and 2) were noted to have echogenicity in the left atrium (LA) [9], and one patient (Case 11) had pulmonary valve stenosis [11]. There is no mention of antenatal EFE in the other three cases (Cases 5, 6, and 10). CHB was present in only 4 (33%) [9,11,17] of the 12 patients (Cases 1, 2, 4, and 11). This was noted at 20 weeks in one patient, 21 weeks in two patients, which includes Case 1, and 24 weeks in one patient, while the changes in EFE were also simultaneously present (Figure 10b). Prior to valve rupture, four patients were noted to have pleural/pericardial effusion associated with severe TR antenatally without reduction in right or left ejection fraction [11,12,14].

### 4.3. Maternal Therapy

Maternal therapy was initiated in seven (58%) patients (Cases 1–4, 9, 11, and 12) [9,10,11,12,17] with EFE changes in the fetal echocardiogram starting from 20 weeks (Case 11) [11] to 34 weeks of gestation (Case 2) [9], except for one case reported by Cuneo et al. (Case 3) [10]. In this case, the therapy was started at 17 weeks of gestation due to a previously affected baby. It is worth noting that the fetus did not have any autoimmune manifestations at 17 weeks but subsequently was noted to have EFE at 19 weeks despite therapy. Dexamethasone was given to six mothers (Cases 1, 2, 9, and 10) [9,10,11,12], while prednisolone was used in one mother (Case 4) [17]. Three (25%) mothers with affected fetuses received intravenous immunoglobulin (IVIG) at 19 weeks (Case 3) [10], 23 weeks (Case 12), and 34 weeks (Case 2) [9] (Figure 11). Three mothers received terbutaline starting at 24 weeks of gestation for CHB (Cases 1, 4, and 11) [11,17]. Three patients (25%) had subsequent progression of the EFE changes during pregnancy despite therapy (Cases 3, 4, and 9) [10,12,17] and, in two, the changes remained stable (Cases 1 and 2) [9]. The effect of therapy is not mentioned in the other two studies [9,11], but they developed valve rupture eventually (Table 2). 

### 4.4. Age of Valve Rupture and Valves Involved

Antenatal valve rupture was noted in three patients, of whom two patients had an isolated TV rupture [8,14]. The third patient had a TV rupture antenatally and a MV rupture postnatally [9]. Two patients were noted to have had a TV rupture immediately post-delivery [12,16]. All MV ruptures were postnatal and occurred from 3 weeks to 5 months of age (Figure 12).

A total of 4 patients (33%) out of the 12 described (Cases 1–4) had both TV and MV valves rupture [9,17]. In Case 2, the TV rupture was noted at birth, while the MV rupture was noted later at 23 days of age [9]. Case 3 had a history of a previously affected pregnancy with autoantibody-mediated CHB and was started on dexamethasone as early as 17 weeks when there were no changes of EFE or CHB. At 19 weeks imaging, patchy echogenicity of the atrioventricular (AV) valves and right ventricle, without AV insufficiency, was observed. Although this patient did not have heart block, a magnetograph showed T wave alternans. She received IVIG at 19 weeks and this was noted to have improved the echogenicity. This infant was delivered at 35 weeks of gestation and received tapering doses of prednisolone for 4 weeks. The neonate then presented with a ruptured valve at 7 weeks of age and underwent valve repair. The baby was on diuretics and ACE inhibitors and had first degree AV block upon follow-up at 6 months of age [10]. In Case 4, tricuspid rupture was noted postnatally within 24 h after pacing with endocardial leads. In this patient, the endocardial leads were removed, and the patient had epicardial leads placed with TV repair. However, the infant developed a MV rupture at 2.5 months of age, requiring MV replacement, and the patient was alive and well as of the report [17]. 

Isolated MV rupture was noted in two (17%) of the patients (Cases 5 and 6) [10,13,15]. Both patients had a valve rupture postnatally. Case 5, as reported in the series by Shiraishi et al., presented at 5 months of age and underwent surgical repair. The exact course of this patient’s illness and outcome, however, are not described in detail [15]. Case 6, with an isolated MV rupture, was noted at 21 days of age, underwent valve repair, and was doing well at 2 years of age [13].

Isolated TV rupture was noted in six patients (50%) (Cases 7–10) [8,9,11,12,14,16]. A TV rupture in Cases 7, 9, and 10 occurred at 39 weeks [8], 34 weeks [12], and 36 weeks [14] of gestation, respectively, which resulted in an emergency cesarean section. In Case 9, there was right-sided failure with worsening TR from 21 weeks of gestation despite dexamethasone therapy. The baby received IVIG and hydrocortisone postnatally, and TV repair was carried out at 2 months of age, and the baby is doing well at 20 months of age [12]. Case 7 and Case 10 underwent TV repair within the first month [8] and at day two of life [14], and were reported to be doing well at eight and two years of age, respectively. TR was noted immediately post-delivery in Case 8 at 38 weeks, while this was absent in the antenatal exams. This infant was repaired at 11 months of age and was doing well at 15 months of age [16]. A postnatal TV rupture was noted in two patients. Case 11 presented at 3 months of age in a decompensated state requiring extracorporeal membrane oxygenation (ECMO). He also had pulmonary valve stenosis and underwent balloon pulmonary dilatation while on ECMO with no improvement, and subsequently a TV repair [11]. Case 12, with an isolated TV rupture, presented at 6 months of age and underwent chordal replacement [9].

### 4.5. Postnatal Management

Postnatal steroids were given to three babies; one received hydrocortisone (Case 9) [12] and the other two received prednisolone for at least four weeks (Cases 2 and 3) [9,10]. Two babies received postnatal IVIG (Cases 2 and 10) [9,14]. Three of the four patients with CHB received an epicardial pacemaker (Cases 1, 4, and 11) [11,17].

The surgical findings in these patients were, predominantly, a disruption of the chordae tendineae just proximal to the insertion into the papillary muscle (Table 3). In four (Cases 7, 10, 11, and 12) [8,9,11,14] of the eight patients [8,9,11,12,14,16] with TV rupture (includes patients with both AV valve rupture), the chordal attachment of the anterior leaflet was disrupted. Cases 8 and 12 had avulsion of the chordae attached to the anterior and posterior leaflets [9,16] and Case 4 had ruptured chordae attached to the lateral and mural leaflets [17]. The tissue of the papillary muscles was noted to be calcified in one patient and frail and fibrotic in another. Patchy fibrosis of both the right ventricular endocardium and the papillary muscle tip is reported in one patient (Case 11) [11]. In cases of MV rupture, the papillary muscles were noted to be pale and atrophic, with laceration of the chordae tendineae attached to the posterior papillary muscle [13,17]. The surgery for valve repair was neochordal attachment with or without annuloplasty in five patients [8,9,11,12,16] and direct anastomosis of chordae to papillary muscle in the other five patients. Only one case (Case 4) underwent MV replacement [17]. Both the MV and TV ruptured in the two patients who died, while infants with single affected valves survived (Figure 13) [9].

### 4.6. Histopathological Findings 

Histopathological findings are available for only three cases with chordal rupture, including Case 1 [9,10]. Cuneo et al. (Case 12) observed severe atrophy with near total replacement of the myocytes by fibrosis and dystrophic calcifications of the papillary muscle with rupture of the AV valve [9]. In Case 3, the surgically excised tissue during valve repair did not show any abnormal findings on histology or immunological staining [10].

## 5. Discussion

### 5.1. Causes of Valve Rupture

Isolated atrioventricular valve insufficiency in the perinatal period with normal valve morphology and without any congenital malformation is rarely reported [18]. Potential etiologies of flail TV in the neonate include congenital endocarditis, ischemia due to premature ductal closure, maternal autoantibody exposure, birth asphyxia, thromboembolism, or rupture due to trauma or during labor [18,19]. Acute MR due to chordal rupture in infancy represents a distinct disease entity, compared to the chordal rupture that is frequently seen in adults due to ischemia or inflammation [15,20]. Valvular regurgitation and rupture in autoimmune diseases in adults are known [21]. Valvar disease due to dysfunction of the tensor apparatus is a severe complication reported in 1.6% of autoimmune antibody-exposed pregnancies with CHB and 4% of non-CHB cardiac patients [1]. The Japanese survey identified 95 cases of MV rupture in infants, of which only 2 cases were attributable to neonatal lupus, indicating the rarity of MV rupture in neonatal lupus patients [15]. In our review of the medical literature, out of 500 articles on lupus in neonates, only 11 case reports of valve rupture have been reported. 

### 5.2. Timing of Valve Rupture

Tricuspid and MV rupture in autoimmune pregnancies are uncommon, with the exception of our case and three other cases that have been documented in the literature [9,10,17]. The valve ruptures were sequential in two cases (Cases 2 and 4), with TV rupture occurring earlier, and simultaneous rupture in the remaining two cases (Cases 1 and 2). The literature indicates that the third trimester or the immediate postpartum period is when most TV ruptures take place. On the other hand, the mean age of presentation for MV rupture is 4 months [20]. Our experience supports the observation that in autoantibody-exposed pregnancies, tricuspid valve rupture occurs perinatally. On the contrary, MV rupture occurred postnatally between 23 days and 5 months of age.

### 5.3. Physiology of Valve Rupture

Severe TV insufficiency in the fetus is poorly tolerated and is associated with a mortality rate as high as 83%, probably because the right heart dominates the antenatal circulation. Ischemia can affect the TV anterior papillary muscle due to its high demand for oxygen, impaired diastolic coronary perfusion during high right ventricular pressure, and location at the distal end of the coronary circulation [18]. Furthermore, it is significant to note that fetal heart failure caused by tricuspid insufficiency is associated with decreased left ventricular function as a result of diastolic dysfunction, highlighting the significance of ventriculo–ventricular interactions [19]. The MV is predominantly experiencing a lower pressure in utero and then converts to a systemic pressure postnatally. Hence, it is possible that a weakened valve apparatus due to long-standing inflammation in utero has resulted in a vulnerable valve postnatally. Moreover, acute MR in neonates is poorly tolerated and results in acute systemic decompensation as the LV volumes are relatively small and diastolic adaptability is limited. In this review, in Cases 3–6, mitral valve ruptures were repaired immediately. However, two mortalities had MV ruptures, which resulted in multisystem dysfunction and precluded repair. It is interesting to note that two cases (Cases 8 and 9) could tolerate a period of medical management before surgery was performed, and, in four cases, the tricuspid valve was repaired immediately.

### 5.4. Endocardial Fibroelastosis

EFE is a non-specific, protracted response to myocardial wall stress that frequently becomes worse over time and eventually leads to heart failure [2]. EFE is one of the extranodal manifestations of neonatal lupus, whose natural history and pathogenesis are less understood [3,4,22,23]. Brito-Zeron et al. analyzed the primary characteristics of 116 autoimmune EFE cases included in 14 investigations, of which nearly 20% had EFE diagnoses without CHB and 7% had diagnoses with CHB [1]. Similar incidence is reported in the review by Qu Y et al. Despite pacemaker therapy, the prognosis for fetuses and newborns with diffuse cardiomyopathy, or EFE, with or without clinical conduction abnormalities, is typically poor, with mortality or the need for heart transplantation occurring in 85% of cases [24]. EFE-associated mortality is also well described in the literature. Jaeggi et al. observed that, although only 11% of autoantibody-induced CHB had clinically significant and echocardiographically detectable EFE, it was associated with 80% of deaths in their cohort [25]. The outcome with babies with EFE was detailed in a study of 103 cases by Izmirly et al., in which there were 51% fatalities; however, the mortality rate of concomitant cardiomyopathy was 100% [3]. Despite sufficient ventricular pacing, Nield et al. found that congestive heart failure returned in 50% of their CHB patients who had developed EFE after a period of clinical stability. This indicates that the changes were unrelated to pacing [4]. In contrast, Qu Y et al. also reported EFE without CHB in nine cases, of which three showed improvement after dexamethasone and three improved spontaneously [24,26].

Both of our cases were referred for evaluation of fetal bradycardia, with CHB and EFE changes noted during the first echocardiographic evaluation but associated with normal cardiac ejection fraction (EF). In this review, 75% (nine cases) of the cases with EFE were detected antenatally, 66% had normal cardiac function, whilst in one patient (Case 9), the EFE changes were noted as early as 19 weeks along with right ventricular dysfunction. 33% (four cases) of the cases with CHB had normal cardiac function. 25% (three of the seven cases) who received maternal therapy had worsening EFE changes but were alive at the time of publication after surgical valve repair. Two patients, including one with CHB who died, had EFE along with rupture of both atrioventricular valves but had received maternal therapy. This suggests that EFE with valve rupture, even without associated CHB or changes in ventricular geometry, could be independently associated with mortality. These observations suggest that CHB may not always be associated with EFE, but valve rupture could develop because of endomyocardial disruptions, which may be genetically driven due to variation in signaling pathway elements [3].

### 5.5. Echocardiography Correlation

Patchy echogenicity on the endocardial surfaces of the fetal heart is one of the prenatal echocardiographic indicators of EFE [1]. Echogenicity was defined as a bright, white-appearing endocardium with well-defined margins. This can be found on the atrial, ventricular lateral wall, septum, or valvar apparatus, and involves varying depths of myocardium [4,22,26]. Llanos et al. described the anatomical and pathological correlations of 18 fetuses with neonatal lupus. Only 3 of the 11 patients with pathologically demonstrated EFE had evidence of echocardiographically demonstrated changes. Valvar changes in the form of fibrosis and calcification were observed postmortem in six patients with heart block; however, an echocardiogram detected a structural valve problem in only one patient. As a result, EFE-related ventricular changes or valvar abnormalities that were clearly demonstrable by histological evaluation may not be observed in echocardiographic measures [27]. EFE may also go unrecognized in some patients with fetal demise or due to a less severe spectrum of the disease [4].

In our review, 75% (9 cases) of the 12 patients have described the antenatal echocardiography findings in the form of patchy echogenicity of the chordae or papillary muscle, similar to our experience. In the other three cases, there was no mention of the antenatal course. There is no diffuse involvement of the endocardium, which is found in other causes of EFE. Since echocardiography is operator-dependent, either there were no changes of EFE, or it was difficult to interpret these findings unless this was specifically looked for. In our Case B, we observed patchy echogenicity of the papillary muscle tips, which was confirmed as fibrosis by histopathology. However, there was no evidence of mitral regurgitation. Hence, patchy echogenicity of the myocardium, especially in the chordae and papillary muscle, taken together with anti-SSA/Ro-SSB/La antibodies in mothers, warrants a close follow-up of evolving valve dysfunction. Anticipatory evaluations toward the assessment of rupture of both atrioventricular valves should be performed.

### 5.6. Histologic Correlation

The progression of EFE that has been documented throughout fetal life and in the first few months of infancy may be explained by the fetal and neonatal autoimmune reactions in response to the maternal autoantibody deposition. Pathologically, it is distinguished by a proliferation of collagen and elastic fiber-rich connective tissue within the endocardium of the cardiac chambers. Most fibroblasts in EFE are proposed to originate from embryonic epicardial-derived mesenchymal cells. A simultaneous effect on the endocardium is thought to occur during infancy when compensatory mechanisms are initiated to counteract cardiac dysfunction brought on by myocyte injury, which encourages fibro-elastotic growth [2].

So far, only Cuneo et al. have described the detailed histological examination in Case 12 with TV rupture, apart from our histological study presented in this review [9]. The papillary muscles showed severe atrophy, with fibrosis and dystrophic calcifications nearly completely replacing the myocytes. The remnant myocytes exhibited compensatory hypertrophic traits, such as abnormally expanded nuclei and more sarcoplasm. Apoptosis was barely detectable by immunological staining. The vascular or perivascular compartments of the papillary muscles did not contain any acute inflammatory infiltrates. In contrast, Neild et al. studied histopathology and immunological staining for three cases with EFE without CHB or valve rupture and observed no evidence of apoptosis, thereby excluding programmed cell death [22]. Of the 18 cases of NL autopsy described by Llanos et al., valve involvement was noted in 6 patients, and EFE changes were noted in 11 patients on autopsy. It is interesting to note that 4 patients with valve involvement and 6 patients with EFE were noted in deaths occurring in the third trimester. In two cases, Llanos et al. observed pancarditis with a mononuclear infiltrate in the endocardium, pericardium, and myocardium, one of which was third-trimester death and the other postnatal [27].

In our experience, both patients studied had histological changes similar to those described above, but only Case B showed signs of inflammatory changes. Giant cells and inflammatory changes in the myocardium, along with calcification and fibrosis of papillary muscles, were seen as early as 21 weeks of gestation in Case B. This suggests that fibrosis and calcification ensue rapidly after the start of inflammation in utero in the early second trimester, with no evidence of cardiac dysfunction or valve regurgitation on an echocardiogram. Numerous studies on animals, including murine, rat, guinea pig, and rabbit models and fetal hearts, have demonstrated that anti-Ro/SSA antibodies from mothers’ sera cause CHB by transplacental passage as early as 11 weeks of gestation [28]. Various immunohistochemistry data speculate that this leads to inhibition of the calcium channels of the fetal heart or apoptotic cell accumulation that induces macrophage infiltration and the release of cytokines, which cause fibrosis and calcification [29]. Increasing and heterogeneous interferon responses, a variety of cell types in the CHB heart, and gene expression enriched in extracellular matrix architecture and fibrosis production have all been discovered recently by single-cell RNA-sequencing [30]. Fetal genetic and maternal factors may play a role in the involvement of a selected fetus with NL [24].

### 5.7. Surgical Repair

In this review, all patients with a ruptured valve underwent surgical repair, except for those who died. It appears that rupture of both valves in proximity of time is poorly tolerated when surgical management also becomes challenging. Despite the changes of EFE, irrespective of postnatal immunotherapy, 10 of the 12 reviewed cases survived after surgical management. Patients with isolated tricuspid or mitral regurgitation due to the rupture of papillary muscles or their chordae or those that rupture sequentially constitute, in some respects, a favorable group for surgical management. Such patients who tolerate surgical repair well have competent pulmonary and aortic valves without left-sided obstructive lesions, well-developed right and left ventricular cavities, and no obstruction of the right ventricular outflow tract. If recognized before the occurrence of irreversible end-organ damage and with proper surgical repair, these children have an excellent chance of a good outcome. If an in utero papillary muscle rupture is diagnosed, the fetus should be carefully monitored until its viability is established, and it should be delivered at the first sign of fetal distress. Before hemodynamic instability worsens, the newborn should be similarly monitored, and surgery should be advised [18]. Hence, based on this review, early-onset antenatal and neonatal significant atrioventricular valve regurgitation (especially at the mitral valve) should be addressed expediently.

### 5.8. Medical Management and Follow-up

Given that circulating maternal autoantibodies may result in EFE, both the mother and the afflicted newborns may benefit from treatment with immunosuppressive drugs, such as corticosteroids, intravenous immune globulin, and plasmapheresis [29]. If fluorinated corticosteroids are used, anti-inflammatory medications administered to the mother may help to lessen endomyocardial inflammation and damage while also lowering the level of circulating autoantibodies [4]. Recent research by Mawad et al. shows that, when compared to previously reported experiences of others, systematic administration of steroids not only increased the survival of fetuses with cardiac NLE but also decreased the postnatal prevalence of DCM. Additionally, they pointed out that postnatal DCM is not specifically correlated with ventricular pacing but rather with the intensity of the inflammatory insult to the fetal myocardium [7]. Some authors have described the use of immunomodulation with IVIG and anti-inflammatory therapies postnatally for valve-associated disease. However, we did not attempt to use any such immunomodulation in Case A. This review also supports the lack of a significant impact of immunomodulators in the prevention of valve rupture.

According to Hedlund et al., neonatal infants delivered to mothers who tested positive for anti-Ro/La but were not treated with immunomodulatory drugs did not exhibit the interferon signature [31]. It has been proposed that maternal hydroxychloroquine treatment can prevent CHB and may provide a molecular basis for such a potential protective effect [32]. We had used hydroxychloroquine in Case A at 24 weeks as well and attempted usage in Case B, but the mother refused to give her consent. We did not find information regarding the use of hydroxychloroquine in other cases in this review.

Zuppa et al. observed the persistence of maternal antibodies in 10% of the offspring until 9 months of age [33]. However, the antibodies’ effects persist long after they are cleared from the bloodstream. According to our analysis, a valve rupture could occur as late as 6 months. However, more research is needed to better define the duration of follow-up for these patients [29,33].

## 6. Limitations

This retroactive analysis has a lot of flaws. Case reports often do not allow assertions of causality because they do not provide a random sample and are fundamentally anecdotal. There are so few patients with valve rupture that the review’s scope was limited. The review was based on the non-systematic reporting of clinical data and, in some cases, was limited in terms of clinical information. This adds to the selection bias [34]. Even though we ran a thorough search that included citation analysis, some papers only had an abstract available. We also reviewed two conference abstracts but chose not to include them in this review based on our original objective. Despite these drawbacks, the study’s strength is its methodical approach and consistent evaluation of high-quality cases of valve rupture.

## 7. Conclusions

There is sparse epidemiological data on the rupture of the tricuspid or mitral apparatus in fetuses or neonates with maternal autoantibodies. A majority of patients with valve rupture had antenatally detected endocardial fibroelastosis in the valvar apparatus. Considering the time frame from the histological changes noted to the detection of EFE on the echocardiogram and the actual valve rupture, it has been difficult to predict the ideal window of opportunity to intervene. We emphasize that these patients should be closely monitored with serial echocardiograms during the vulnerable perinatal period. Newer modalities, such as cardiac magnetic resonance imaging, may add value in prognosticating the valvar EFE changes. As rupture of both atrioventricular valves occurring at close intervals carries a high mortality risk, possibly treating the disease earlier in the course may prevent further fibrosis; however, we also realize that by the time these findings are observed by imaging, the window has perhaps long passed. We feel that EFE changes in chordae, papillary muscle, or valve leaflets need to be taken seriously, and anti-inflammatory treatment needs to be started promptly to prevent valve rupture. Appropriate and expedited surgical repair of ruptured atrioventricular valves is feasible and has a low mortality risk. An aggressive antenatal imaging protocol for neonatal lupus patients, irrespective of rhythm changes, along with a low threshold for surgical interventions with the early onset of significant valve regurgitation, should be adopted. Such a proactive strategy would detect this minority of patients earlier, enabling them to receive more specialized medical care or to undergo prompt repair.

## Figures and Tables

**Figure 1 diagnostics-13-01481-f001:**
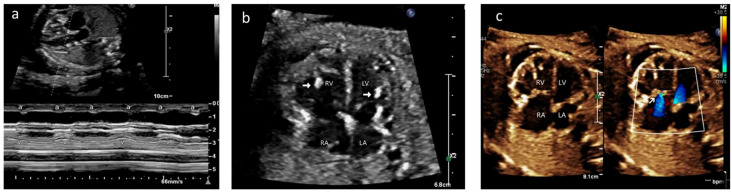
Case A: (**a**) M-mode echocardiogram showing atrioventricular dissociation; ‘a’ shows atrial contractions and ‘V’ shows ventricular contractions. (**b**) Antenatal echocardiogram, four-chamber view, showing hyperechoic papillary muscles (white arrow) in the left ventricle (LV) and right ventricle (RV). RA—right atrium, LA—left atrium. (**c**) Antenatal echocardiogram, four-chamber view, showing mild tricuspid regurgitation (white arrow) and no mitral regurgitation with normal atrioventricular valves.

**Figure 2 diagnostics-13-01481-f002:**
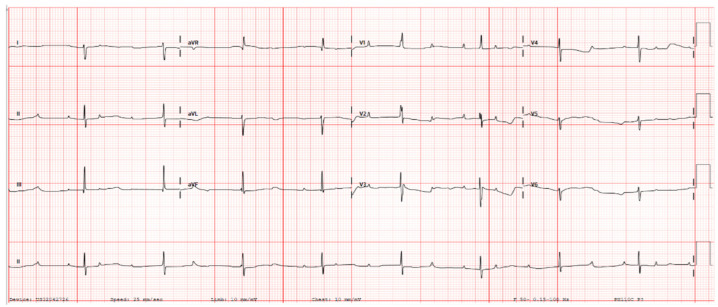
Case A—Postnatal electrocardiogram showing complete heart block. There is complete atrioventricular dissociation, an atrial rate of 125 bpm, and a ventricular rate of 52 bpm.

**Figure 3 diagnostics-13-01481-f003:**
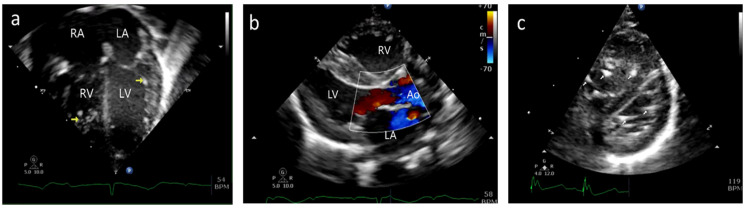
Case A: (**a**) Postnatal echocardiogram—four-chamber view showing normal mitral and tricuspid valves. Hyperechoic papillary muscles (yellow arrow) were noted. RA—right atrium, LA—left atrium, RV—right ventricle, LV—left ventricle. (**b**) Immediate postnatal echocardiogram, long axis view, showing no regurgitation of the mitral valve. Ao—Aorta. (**c**) Postnatal echocardiogram, short axis at the level just below the mitral valve, showing hyperechoic papillary muscles in the left (bottom) and right (top) ventricles (white arrow).

**Figure 4 diagnostics-13-01481-f004:**
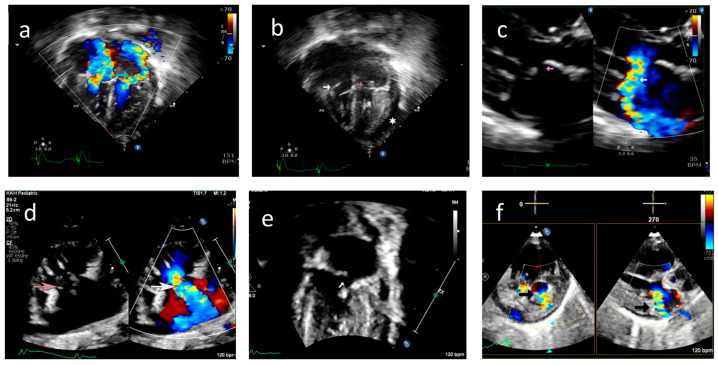
Case A: (**a**) Echocardiogram, four-chamber view with color doppler showing severe tricuspid regurgitation (pink arrow) and severe mitral regurgitation (green arrow) noted on day 45 of life. (**b**) Echocardiogram, four-chamber view, demonstrating ruptured chordae resulting in prolapse of tricuspid valve leaflets (white arrow) and mitral valve leaflets (pink arrow). *—Pericardial effusion. (**c**) 2D and color doppler long axis views of the mitral valve showing valve prolapse (pink arrow in 2D image) and severe mitral valve regurgitation (white arrow in color image) (**d**) Modified long axis view on a 2D echocardiogram, showing rupture of the tricuspid valve chordae (pink arrow) and severe tricuspid regurgitation (white arrow). (**e**) Echocardiogram, four-chamber view of ruptured mitral valve leaflet chordae causing mitral valve prolapse (white arrow). (**f**) X-plane color doppler image on a 2D echocardiogram showing severe mitral regurgitation (black arrow).

**Figure 5 diagnostics-13-01481-f005:**
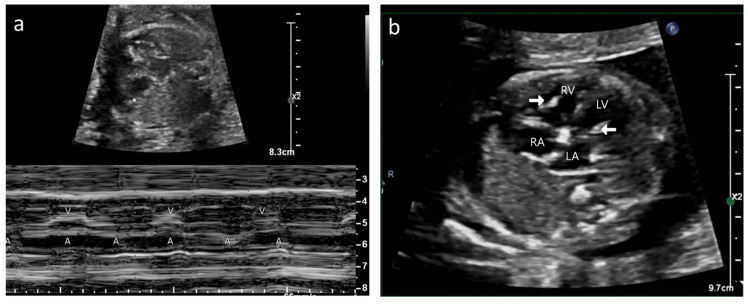
Case B: (**a**) M-mode echocardiogram showing atrioventricular dissociation in the fetus. A—atrial contractions; V—ventricular contractions. (**b**) Echocardiogram, four-chamber view—hyperechoic papillary muscle (white arrow in RV and LV).

**Figure 6 diagnostics-13-01481-f006:**
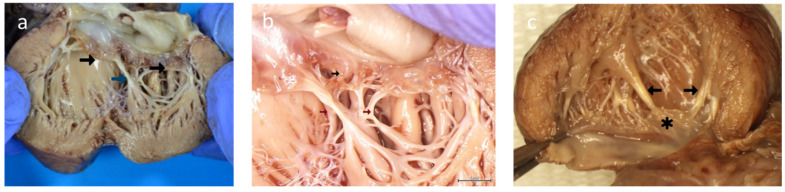
(**a**) Case A—postmortem gross morphology: mitral valve with ruptured chordae of the anterior leaflet of the mitral valve resulting in shrunken leaflets (black arrow). Calcified chordae (blue arrow). (**b**) Case A—postmortem gross morphology: mitral valve with retraction and loss of leaflet tissue (black arrow). Calcified/fibrotic chordae (brown arrow). (**c**) Case B—postmortem gross morphology: left ventricle showing normal mitral valve leaflets (*). Fibrotic/calcified chordae and papillary muscle tips (black arrows).

**Figure 7 diagnostics-13-01481-f007:**
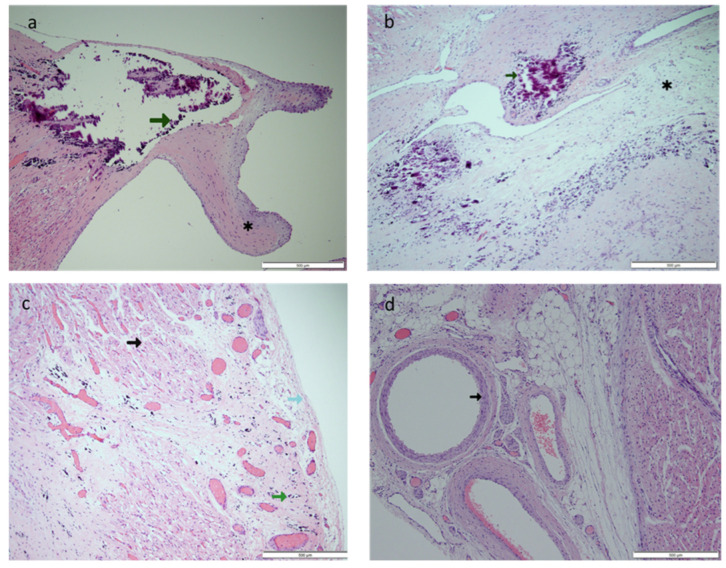
Case A: (**a**) Hematoxylin and eosin-stained section (original magnification 4×) of the left ventricular papillary muscle with calcified tip (green arrow). *—ruptured chordae showing fibrosis. (**b**) Hematoxylin and eosin-stained section (original magnification 10×) of the atrioventricular node area with calcification (green arrow) and fibrosis (*). (**c**) Hematoxylin and eosin-stained section (original magnification 20×) of the myocardium (black arrow) and pericardium (light blue arrow) showing extensive subpericardial calcification (green arrow) extending to the myocardium. (**d**) Hematoxylin and eosin-stained section (original magnification 10×) of the coronary artery (black arrow), which showed no inflammation.

**Figure 8 diagnostics-13-01481-f008:**
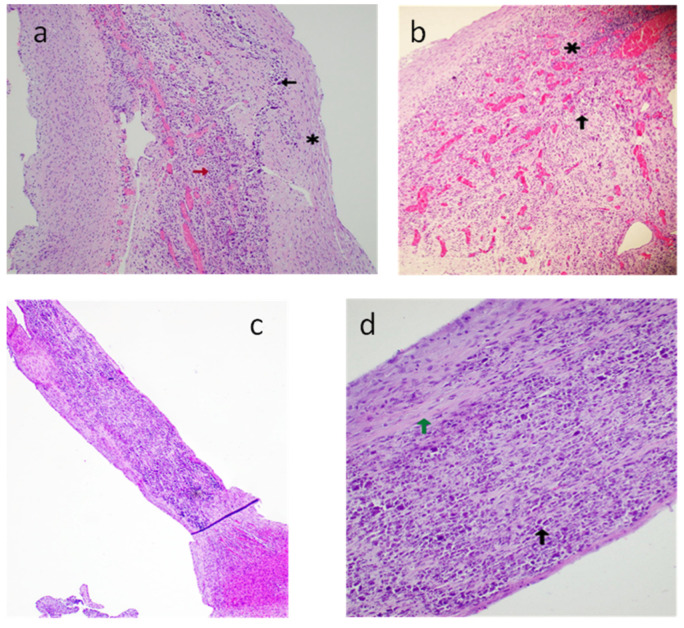
Case B: (**a**) Hematoxylin and eosin-stained section of the atrioventricular node (original magnification 10×)—calcification (black arrow), fibrosis (*), granulation tissue with neovascularization (red arrow). (**b**) Hematoxylin and eosin-stained section (original magnification 10×) of the atrioventricular node—acute inflammatory changes with granulation tissue (*), multinucleated giant cell (black arrow). (**c**) Hematoxylin and eosin-stained section (original magnification 4×) of the right ventricular papillary muscle with chordae. (**d**) Hematoxylin and eosin-stained section (original magnification 10×) of the right ventricular papillary muscle with chordae, fibrosis (green arrow), and calcification (black arrow).

**Figure 9 diagnostics-13-01481-f009:**
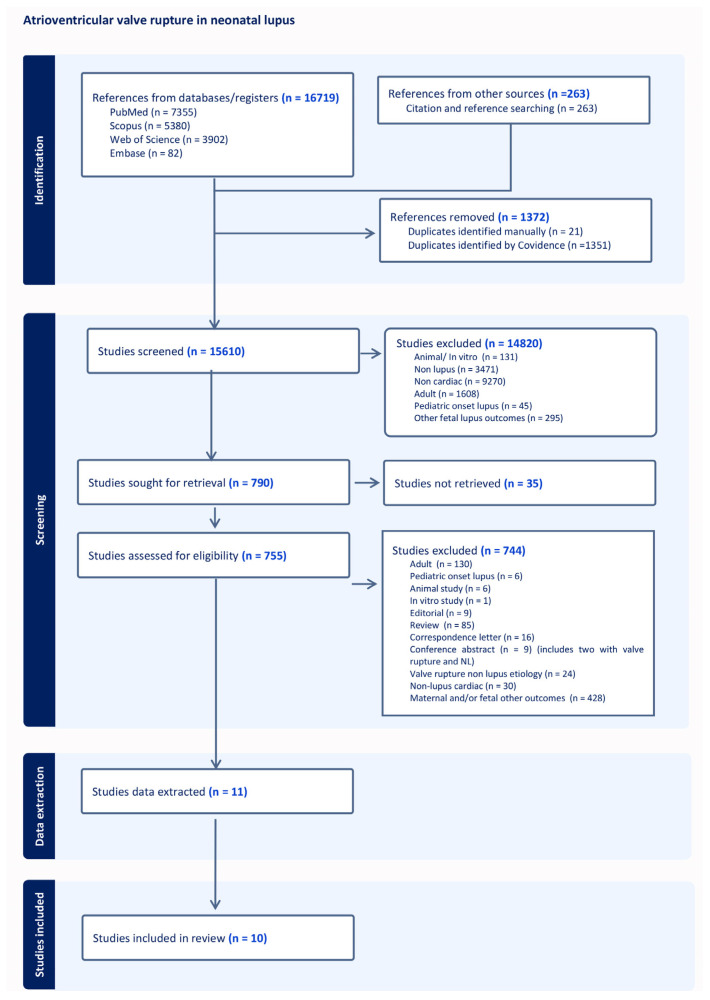
PRISMA flow diagram for systemic review.

**Figure 10 diagnostics-13-01481-f010:**
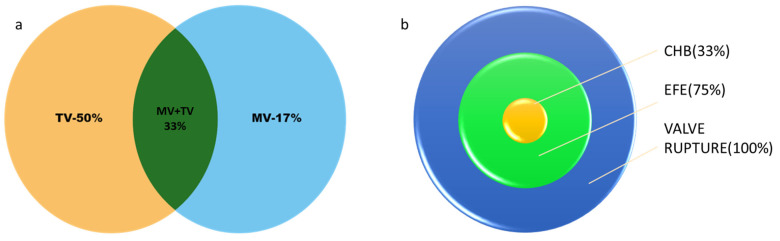
(**a**) Venn diagram showing tricuspid valve rupture (TV) in 50% (*n* = 6), mitral valve rupture (MV) in 17% (*n* = 2), and both mitral and tricuspid valve rupture (MV+TV) in 33% (*n* = 4). (**b**) Target chart showing patients with valve rupture (*n* = 12), of whom 75% (*n* = 9) had changes of endocardial fibroelastosis (EFE) reported and 33% (*n*= 4) had complete heart block (CHB).

**Figure 11 diagnostics-13-01481-f011:**
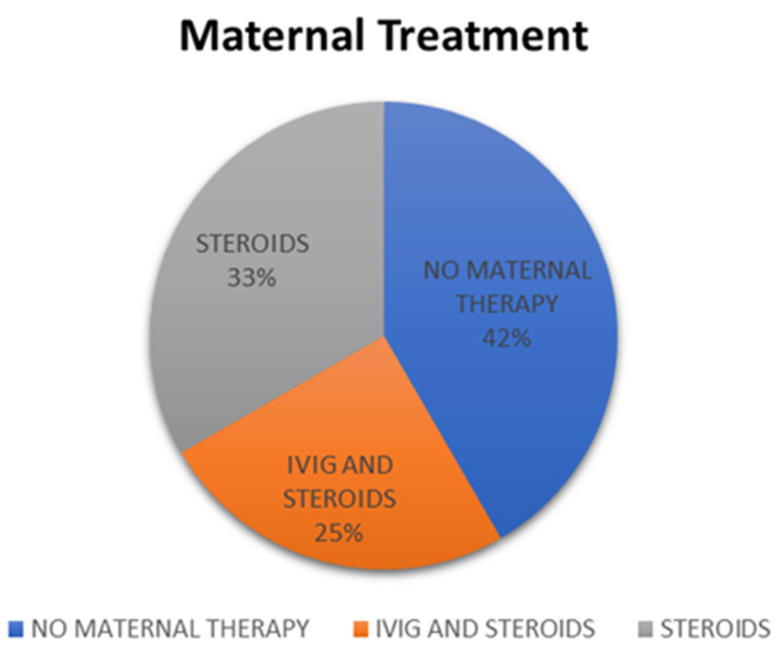
Pie chart showing mothers who received no antenatal therapy (*n* = 5), steroids only (*n* = 4), and intravenous immunoglobulin (IVIG) and steroids (*n* = 3).

**Figure 12 diagnostics-13-01481-f012:**
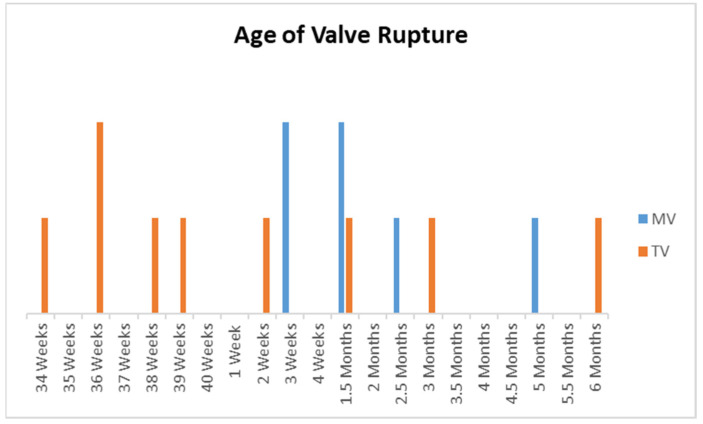
Bar chart showing the timing of rupture for the tricuspid valve (TV) and mitral valve (MV). X-axis: age of valve rupture from 34 to 40 weeks antenatal and subsequently postnatal. Y-axis: number of patients.

**Figure 13 diagnostics-13-01481-f013:**
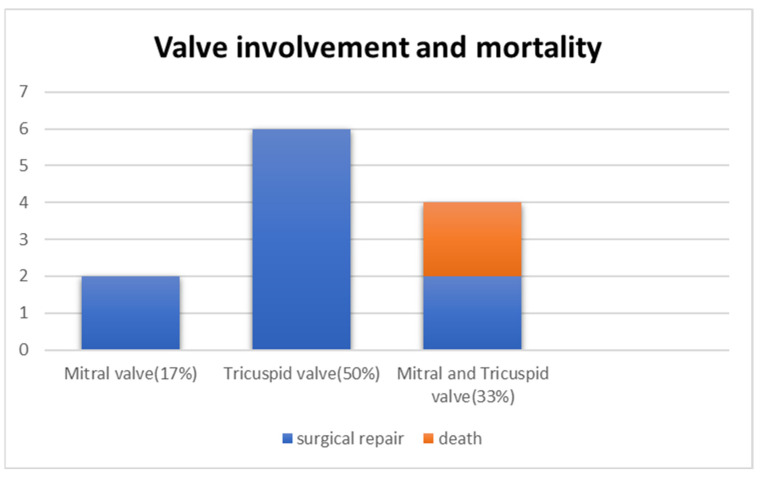
Bar chart showing valve involvement (blue) and mortality (orange). X-axis represents the valve involved. Y-axis shows the number of patients.

**Table 1 diagnostics-13-01481-t001:** Maternal characteristics of the cases with valvar rupture in autoantibody-exposed pregnancy.

Case	Reference Year of Publication	Maternal Age (Years)	Gravida	Prior Affected Pregnancy	Known Case of Auto Immune Disease	Anti-Ro/Anti La
1	Case A	35	2	No	No	+/+
2	Cuneo et al. 2011 [9]	25	2	No	Yes	+/−
3	Cuneo et al. 2009 [10]		2	Yes	Sjogren’s disease	+/−
4	Weber et al. 1994 [17]	31	2	Yes	Sjogren’s disease	+/−
5	Shiraishi et al. 2014 [15]			-	-	
6	Hamaoka et al. 2009 [13]			-	-	+/+
7	Bellon et al. 2022 [8]	26	1	-	No	+/−
8	Tarca et al. 2017 [16]	31	2	1	Antiphospholipid syndrome	+/+
9	Gonzalez-Lopez et al. 2017 [12]		2	-	No	+/+
10	Brooks et al. 2015 [14]		1		No	+/+
11	Fleming et al. 2008 [11]				No	+/−
12	Cuneo et al. 2011 [9]	27	2	0	Yes	+/−

**Table 2 diagnostics-13-01481-t002:** Antenatal echocardiographic features for neonatal lupus with time of identification.

Case	GA (Weeks) for Detection			Valve Involved	Maternal Therapy			Timing of Valve Rupture			GA (Weeks) Delivered
	EFE	CHB	Effusions		Medication	Initiation	Effect	Antenatal	Perinatal	Postnatal	
1	21	21	21–29	MV+TV	D+T	24	Pleural effusions resolved, reduction in pericardial effusion to mild	-	-	TV and MV—45 days	32 + 4
2	19	-	19	MV+TV	D+IVIG	34	No change	TV—34–36 weeks	-	MV—23 days	36
3	19	-	-	MV+TV	D+IVIG	17	Improvement in patchy echogenicity	-	-	TV and MV—7 weeks	35
4	24	24	-	MV+TV	P+T	22	Increased echogenicity along the AV valves without insufficiency, no hydrops fetalis	-	-	TV = 2 weeks MV—2.5 months	35
5			-	MV	-	-	-	-	-	5 months	
6		-	-	MV	-	-	-	-	-	21 days	40
7	20	-	39	TV	-	-	-	39 weeks	-	-	39
8	19	-	-	TV	-	-	-	-	Immediately after delivery	-	38 + 6
9	21	20	-	TV	D	21	Worsening of TR, moderate RV dysfunction	-	Immediately after delivery	-	34
10	-	-	36	TV	-	-	-	36 weeks	-	-	36
11	20	20	-	TV	D+T	20		-	-	3 months	34
12	21	21	-	TV	D+IVIG	23	No change	-	-	6 months	38

GA: gestational age in weeks, EFE: endocardial fibroelastosis, CHB: complete heart block, MV: mitral valve, TV: tricuspid valve, AV: atrioventricular, RV: right ventricle, IVS: interventricular septum, D: dexamethasone, P: prednisolone, IVIG: intravenous immunoglobulin, T: terbutaline

**Table 3 diagnostics-13-01481-t003:** Postnatal management with surgical/postmortem findings of patients with valve rupture and their outcomes.

Case	Postnatal Therapy	Age of Rupture	Age of Surgery	Surgery	Outcome
1	-	45 days	-	Not performed	Died at 55 days of life
2	-	TV—34–36 weeks MV—23 days	-	Not performed	Died at 32 days
3	Tapering prednisolone over 4 weeks	7 weeks	7 weeks	Repair of mitral and TV chordae with gortex and autologous pericardium	6 months old—diuretics, ACE inhibitors, first degree AV block, and incomplete RBBB
4	-	2 weeks—TV after placing endocardial lead, MV—2.5 months	2 weeks VVI, 24 h later TV repair and epicardial pacemaker; 2.5 months—MV replacement	1. Anastomosis of the TV chordae to the papillary muscle and atrial septum was closed 2. MV replacement	Doing well
5	-	5 months		-	-
6	-	21 days	21 days	Annuloplasty and anastomosis of chordae with autologous pericardium	2 year—1st degree heat block
7	-	Antenatal	Neonate	Direct attachment of the chordae to neo-papillary muscle and TV annuloplasty	Well at 8 years
8	-	Delivery	11 months	Neochordae with gortex attached to septum and annuloplasty	Well at 15 months
9	Hydrocortisone, IVIG, captopril, sildenafil	Delivery	2 months	Polytetrafluoroethylene neochordae, No annuloplasty	20 months
10	-	36 weeks	6 days	TV repair	Normal at 2 years
11	-	3 months	Permanent pacemaker: 3 days, BPV: 3 months, TV repair at 3 months	Repair of anterior tricuspid leaflet with PTFE cords, closure of ASD, dilatation of pulmonary valve	Well
12	Prednisolone for 2 months, IVIG 2 g/kg	6 months	6 months	Repair with Gore-Tex chordae	Well at 3 years

TV: tricuspid valve, MV: mitral valve, IVIG: intravenous immunoglobulin, ACE: angiotensin converting enzyme, RBBB: right bundle branch block.

## Data Availability

The data presented in this study are available in the article.

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
