# Peer review of "Endocardial Fibroelastosis as an Independent Predictor of Atrioventricular Valve Rupture in Maternal Autoimmune Antibody Exposed Fetus: A Systematic Review with Clinicopathologic Analysis"

_diagnostics, 2023, doi:10.3390/diagnostics13081481_

Round 1

Reviewer 1 Report

It is a well written and interesting paper. 

The authors describe in details the cases and support everything with enough tables, figures, video clips etc.. for a better visualization. 

LE is very common in pregnant women and this papers also contribute well for clinical assessment 

Author Response

Dear Reviewer,

We acknowledge your response.

Thank you for your comments.

Yours Sincerely,

Dr Sreekanthan Sundararaghavan

Reviewer 2 Report

The manuscript describes and investigates a case study of atrioventricular valve rupture as a result of fetal exposure to maternal autoantibodies. The work illustrates the case and compares it, from a histopathological perspective, with a case of abortion due to a congenital complete heart block, as one of the most widespread manifestations of neonatal lupus. Finally, a thorough review of the literature on atrioventricular valve ruptures in neonatal lupus is carried out to further support and discuss the case study findings.

While the manuscript is well written, the organization and layout of the work should be carefully revised and some contents should be clarified.

Detailed comments/suggestions are reported below:

ABSTRACT:

- An abstract should at least mention some of the major findings of the study and briefly report the main conclusions of the work. Please consider including this information in the Abstract of the present work.

1. INTRODUCTION:

- The Introduction is clear and straightforward. Despite that, it appears to be a bit too concise. I would suggest adding further preliminary information and context before introducing the case studies to the reader.

2. CASE PRESENTATION

- "Paragraph 2.3.3. Review of the literature": this part of the work should be moved to a separate section (e.g. Section 3. Literature review).

- 2.3.3: it would be worth presenting a flow chart that systematically describes/summarizes the literature search (how many articles included, how many excluded, etc...). As an example, you could follow a typical PRISMA flow diagram model (https://guides.lib.unc.edu/prisma).

3. DISCUSSION

- Sections 3.6, 3.7, and 3.8 report a mere list of findings form the reviewed articles. Further interpretations, contextualization, and indications should be given by the authors based on the examined literature.

4. CONCLUSION

- The Conclusions are not adequately supported by the results. Please consider revising this section by highlighting the major findings that emerged from both the case studies presented and the literature review carried out.

- What can the authors really say after investigating the presented case studies in light of the literature review they carried out? This point is probably missing in the Conclusion section.

- "...Such an approach would ideally identify this subgroup of patients earlier...", the authors mention an approach that has not been adequately and systematically described. Further details, procedures, information, and structure of the proposed approach should be provided if the authors aim to suggest/propose an approach for the early detection of valve ruptures in neonatal lupus.

OTHER COMMENTS:

- I would recommend a careful revision of the punctuation and figure formatting.

- Please check sections' titles and text formatting (e.g. alignment of section 2.3.3 and of section 3.1), font type (e.g. some titles are in upper-case while some others are lower-case), font size (e.g. sections 3.6, 3.7. and 3.8 have smaller or larger font size), and font color (some parts of section 3.8 are in blue color).

- Table captions should be reported above the Table itself, not below (e.g. check Tabel 2 and Table 3).

- Minor English spelling errors should also be checked.

- Please carefully check MDPI formatting guidelines.

Reviewer 3 Report

This paper presented a clinicopathologic analysis and literature review of endocardial fibroelastosis as an independent predictor of AV valve rupture. The paper is well-written and organized. To be honest I didn't have experience in reviewing case report papers. However, I feel that the data and conclusions drawn from the cases are very important for the researchers in this area. Therefore I support the publication of this manuscript.

Author Response

(The authors gave the same response as above.)

Round 2

Reviewer 2 Report

The authors have addressed previous comments and suggestion and the manuscript has been significantly improved. 
I would recommend detailing exclusion reasons in the PRISMA flow diagram for the first box on the left in the Screening section. 

Author Response

Thank you for your Comments and Suggestions.

The authors have addressed previous comments and suggestion and the manuscript has been significantly improved. 
I would recommend detailing exclusion reasons in the PRISMA flow diagram for the first box on the left in the Screening section.

Response:

We have amended the PRISMA flow diagram as suggested.

Yours Sincerely,

Dr Sreekanthan Sundararaghavan.